# OpenReview forum: "Reliable and Efficient Amortized Model-based Evaluation"
_ICML.cc/2025/Conference — ICML 2025 poster_

### Official Review · Reviewer_Kyh8 · 2025-03-06

**Overall Recommendation:** 3

**Summary:**

This Paper proposes a new approach to evaluate LLM performance via IRT and to provide item generation with pre-chosen difficulty levels. I am not an expert in LLMs  but I happened to work on IRT in the past. So my point mainly concern this aspect.

I am very short on time for ICML reviews. Apologies for my reviews being a bit short.

**Claims And Evidence:**

The authors claims that their procedure simplifies evaluation of LLM-based models and even enables to generate new test items on the fly. As far as I understand it, the experiments support both claims.

**Essential References Not Discussed:**

see above

**Experimental Designs Or Analyses:**

I am not an expert in LLM evaluation but the general approach and metrics looked sensible to me.

**Methods And Evaluation Criteria:**

I am not an expert in LLM evaluation but the general approach and metrics looked sensible to me.

**Other Comments Or Suggestions:**

see above

**Other Strengths And Weaknesses:**

For the Rasch model, the number of correctly answered item by a person / the number of people correctly answering an item are sufficient statistics for the person / item parameters, respectively. This means that for the Rasch model, EM etc. algorithms are not really necessary since the item and person parameters are essentially analytic. Is there a reason, this simple solution to the Rasch model was not considered?

Why use only the Rasch model in your paper? The Rasch model is often overly restrictive. Typicilly, at least a 2 parameter logistic (2PL) model is appropriate, at least when working with humans.

I struggled understanding what exactly is done in the paper from the abstract. Perhaps the author could increase the context of the paper a bit at the start of the abstract?

**Questions For Authors:**

see above

**Relation To Broader Scientific Literature:**

I am not familiar with the LLM literature. In terms of IRT literature, the fact that only the Rasch model was considered is somewhat problematic for me (see below).

**Theoretical Claims:**

None

---

> ### Author Rebuttal · Authors · 2025-03-29
>
> Dear Reviewer Kyh8,
>
> Thank you for your valuable feedback. We answer your comment below.
>
> **Answer to Other Strengths And Weaknesses 1:** When the difficulty (ability) is known, the sum of the responses is indeed a sufficient statistic for ability (difficulty). During calibration, we generally do not know either difficulty or ability. We can’t estimate them jointly because of the incidental parameter problem, first recognized by Neyman and Scott (Neyman-Scott 1948). Harwell (1988) provides a detailed discussion of this problem in IRT. The marginal likelihood approach is then proposed to avoid this problem by marginalizing out the ability during difficulty estimation. This approach is broadly known in the related literature as the “expectation-maximization” algorithm (Harwell 1988). Below, we explore whether the maxima of this marginal likelihood objective can be obtained analytically. Consider one question with difficulty $z$ and a response vector $Y$ size $N$. The log marginal likelihood function is
>
> $$
> \mathcal{L}(z) = \frac{1}{N} \sum_i \log p(Y|z)
> $$
>
> $$
> = \frac{1}{N} \sum_i \mathbb{E}_{\theta_i} \log p(Y_i|\theta_i, z)
> $$
>
> $$
> \approx \frac{1}{N M} \sum_{i,k} \log p(Y_i|\theta_{i,k}, z)
> $$
>
> $$
> = \frac{1}{N M} \sum_{i,k} Y_i \log p_{i,k}(z) + (1- Y_i) \log (1- p_{i,k}(z))
> $$
>
> where $p_{i,k}(z) = \frac{1}{1 + e^{-(\theta_{i,k} - z)}} = \sigma(\theta_{i,k}-z)$. Taking derivative with respect to $z$ using chain rule and $\sigma'(x) = \sigma(x) (1-\sigma(x))$:
>
> $$
> \frac{d\mathcal{L}}{dz} = \frac{1}{N M} \sum_{i,k} \left[ \frac{Y_i}{p_{i,k}(z)} \frac{d p_{i,k}(z)}{dz} - \frac{1 - Y_i}{1 - p_{i,k}(z)} \frac{d p_{i,k}(z)}{dz} \right]
> $$
>
> $$
> = \frac{1}{N M} \sum_{i,k} \left( \left[ \frac{Y_i}{p_{i,k}} - \frac{1 - Y_i}{1 - p_{i,k}} \right] \frac{d p_{i,k}}{dz} \right)
> $$
>
> $$
> =  \frac{1}{N M} \sum_{i,k} \left( Y_i - Y_i p_{i,k} - p_{i,k} + Y_i p_{i,k}\right)
> $$
>
> $$
> =  \frac{1}{N M} \sum_{i,k} Y_i - p_{i,k}
> $$
>
> Setting the derivative to zero:
> $$
> \frac{d\mathcal{L}}{dz} = \frac{1}{N M} \sum_{i,k} \left( Y_i - p_{i,k} \right) = 0   \Rightarrow
> \sum_{i,k} Y_i = \sum_{i,k} \frac{1}{1 + e^{-(\theta_{i,k} - z)}}
> $$
>
> This expression does not admit an analytic solution for $z$ except in degenerative cases because the sum of the logistic function generally does not simplify to a closed-form invertible expression in $z$.
>
> [1] Harwell, Baker, Zwarts. Item Parameter Estimation Via Marginal Maximum Likelihood and an EM Algorithm: A Didactic. Journal of Educational Statistics, 1988, pp. 243-271
>
> [2] Neyman, Scott. Consistent Estimates Based on Partially Consistent Observations. Econometrica, pp. 1-32
>
> **Answer to Other Strengths And Weaknesses 2:** We conducted an ablation study comparing 3 IRT variants—Rasch, 2PL, and 3PL (see Figure 9). The results indicated that neither the 2PL nor the 3PL models outperformed the Rasch model, a finding we attribute to the limited number of test takers in the LLM context. With additional parameters, the more complex models tend to suffer from increased estimation complexity, which in turn raises the risks of overfitting and higher variance. Based on these considerations, we chose the Rasch model.
>
> **Answer to Other Strengths And Weaknesses 3:** We have revised our abstract: Comprehensive evaluations of language models (LM) during both development and deployment phases are necessary because these models possess numerous capabilities (e.g., mathematical reasoning, legal support, or medical diagnostic) as well as safety risks (e.g., racial bias, toxicity, or misinformation). The average score across a wide range of benchmarks provides a signal that helps guide the use of these LMs in practice. Currently, holistic evaluations are costly due to the large volume of benchmark questions, making frequent evaluations impractical. A popular attempt to lower the cost is to compute the average score on a subset of the benchmark. This approach, unfortunately, often renders an unreliable measure of LM performance because the average score is often confounded with the difficulty of the questions in the benchmark subset. Item response theory (IRT) was designed to address this challenge, providing a reliable measurement by careful controlling for question difficulty. Unfortunately, question difficulty is expensive to estimate. Facing this challenge, we train a model that predicts question difficulty from its content, enabling a reliable measurement at a fraction of the cost. In addition, we leverage this difficulty predictor to further improve the evaluation efficiency through training a question generator given a difficulty level. This question generator is essential in adaptive testing, where, instead of using a random subset of the benchmark questions, informative questions are adaptively chosen based on the current estimation of LLM performance. Experiments on 22 common natural language benchmarks and 172 LMs show that this approach is more reliable and efficient compared to current common practice.

---

### Official Review · Reviewer_2sSn · 2025-03-14

**Overall Recommendation:** 4

**Summary:**

The paper introduces a new way to evaluate large language models (LLMs) using Item Response Theory (IRT), a method from psychology that helps measure abilities and item difficulties separately. Traditional evaluation methods can be expensive and depend too much on the specific test questions chosen, so this paper aims to make the process more reliable as well as efficient. It includes two main innovations: (i) Amortized Calibration and (ii) Conditional Item Generator.

**Claims And Evidence:**

The paper makes several key claims:
1. The IRT-based method is more reliable and efficient than traditional Classical Test Theory (CTT) methods. This is backed by Table 1.
2. The conditional item generator creates effective new questions, backed by Sec 4.4

**Essential References Not Discussed:**

Not much. But it would be better to discuss related topics such as LLM performance prediction. The current related work is rather too short.

**Experimental Designs Or Analyses:**

The experimental setup involved 25 NLP datasets (e.g., airbench, mmlu, truthful_qa) and 184 LLMs (e.g., ada, LLaMA, GPT-4). The experiments are solid enough to support the conclusion.

**Methods And Evaluation Criteria:**

This paper measures different approaches with common metrics like AUC-ROC.

**Other Comments Or Suggestions:**

-

**Other Strengths And Weaknesses:**

-

**Questions For Authors:**

-

**Relation To Broader Scientific Literature:**

The work builds on psychometric theory, particularly IRT, with roots in educational testing and extends it to AI evaluation, aligning with recent efforts in scalable and efficient LLM evaluation.

**Theoretical Claims:**

This paper provides certain modifications with theoretical claims. But I am not able to justify whether they are right or wrong.

---

> ### Author Rebuttal · Authors · 2025-03-28
>
> Dear Reviewer 2sSn,
>
> Thank you for your valuable feedback. We answer your comment below.
>
> **Essential References Not Discussed:** Not much. But it would be better to discuss related topics such as LLM performance prediction. The current related work is rather too short.
>
> **Answer:** Thank you for your valuable feedback. We agree that a more detailed discussion on LLM performance prediction would strengthen our related work section. In response, we will incorporate the following content into our updated submission:
>
> Recent research has made significant strides in understanding and predicting LLM performance. For instance, Schaeffer et al. (2023) address performance discontinuities associated with emergent behaviors, while Ganguli et al. (2022a), Owen (2024), and Finnveden (2020) have illustrated how downstream task performance can be systematically predicted. In one study, Hu et al. (2024) established a clear relationship between the amount of training resources and the resulting performance on downstream tasks by iteratively pretraining a model. Moreover, Arora and Goyal (2023) offer insights into forecasting performance by breaking down complex language model capabilities into fundamental skills. Recent work by Ruan et al. (2024) further enhances scaling laws by integrating latent variables that capture underlying patterns across various model families and tasks. These works on predictable model performance are complementary to research in IRT that helps improve the efficiency of model evaluation.
>
> [1] Rylan Schaeffer, Brando Miranda, and Sanmi Koyejo. 2023. Are emergent abilities of large language models a mirage? In Conference on Neural Information Processing Systems.
>
> [2] Deep Ganguli, Danny Hernandez, Liane Lovitt, Amanda Askell, Yuntao Bai, Anna Chen, Tom Conerly, Nova Dassarma, Dawn Drain, Nelson Elhage, Sheer El Showk, Stanislav Fort, Zac Hatfield-Dodds, Tom Henighan, Scott Johnston, Andy Jones, Nicholas Joseph, Jackson Kernian, Shauna Kravec, Ben Mann, Neel Nanda, Kamal Ndousse, Catherine Olsson, Daniela Amodei, Tom Brown, Jared Kaplan, Sam McCandlish, Christopher Olah, Dario Amodei, and Jack Clark. 2022a. Predictability and surprise in large generative models. In Conference on Fairness, Accountability, and Transparency. ACM.
>
> [3] David Owen. 2024. How predictable is language model benchmark performance? In arXiv.
>
> [4] Lukas Finnveden. 2020. Extrapolating gpt-n performance.
>
> [5] Shengding Hu, Xin Liu, Xu Han, Xinrong Zhang, Chaoqun He, Weilin Zhao, Yankai Lin, Ning Ding, Zebin Ou, Guoyang Zeng, Zhiyuan Liu, and Maosong Sun. 2024. Predicting emergent abilities with infinite resolution evaluation. In International Conference on Learning Representations.
>
> [6] Sanjeev Arora and Anirudh Goyal. 2023. A theory for emergence of complex skills in language models. In arXiv.
>
> [7] Yangjun Ruan, Chris J. Maddison, and Tatsunori Hashimoto. 2024. Observational scaling laws and the predictability of language model performance. In arXiv.

---

> > ### Comment · Reviewer_2sSn · 2025-04-03
> >
> > Thanks for the rebuttal. The reviewer has no further comment on this paper except for two points:
> > 1. Some references (Both in the rebuttal and the paper) listed are the ArXiv version, not the proceedings version. Kindly use the proceedings version.
> > 2. The related work on performance prediction (rebuttal) could be further improved for comprehensiveness. It appears the reference mainly comes from ML conferences like ICLR, ICML or NeurIPS. Perhaps keyword searching in NLP conference proceedings like ACL or EMNLP could help? It would be great to recognize contributions of papers from other venues on the related topics.

---

> > > ### Author Response · Authors · 2025-04-08
> > >
> > > Thank you for your comment, we have fixed the citation issue in the following related work on LLM performance prediction, and we will fix the reference list of the paper in the final version. We have added more related work from NLP conferences like ACL and EMNLP.
> > >
> > >
> > > Recent research has significantly advanced our understanding of LLM performance prediction by establishing robust scaling laws and uncovering emergent phenomena. Kaplan et al. (2020), Hoffmann et al. (2022), and Hernandez et al. (2022) laid the groundwork by elucidating how model performance scales with size, data, and compute. Bahri et al. (2024) and Muennighoff et al. (2023) have deepened these insights, while studies such as those by Isik et al. (2024), Ghorbani et al. (2021), Zhuocheng et al. (2023), Caballero et al. (2023), and Henighan et al. (2020) have extended scaling laws to predict downstream task performance. Research on predicting emergent abilities with infinite resolution evaluation (2024) has highlighted the sudden performance gains. Schaeffer et al. (2023) examined discontinuities linked to emergent abilities, while Finnveden (2020) explored methods for extrapolating GPT performance. Ganguli et al. (2022a) and Owen (2024) scrutinized the balance between predictability and surprise in generative models, and Arora and Goyal (2023) broke down complex LLM skills into fundamental components to facilitate granular forecasting. Moreover, studies on emergence phenomena by Suzgun et al. (2022) and Wei et al. (2022) have shed light on the mechanisms behind abrupt performance improvements. Ruan et al. (2024) introduced latent variables that generalize across tasks and model families. Zhang et al. (2024) proposed a collaborative framework that leverages cross-family model-task performance patterns through factor analysis. Finally, to address broader challenges in the field, Anwar et al. (2024) highlighted foundational issues in the alignment and safety of LLMs.
> > >
> > > [1] Rylan Schaeffer et al. 2023. *Are emergent abilities of large language models a mirage?* NeurIPS.
> > >
> > > [2] Deep Ganguli et al. 2022. *Predictability and surprise in large generative models.* FAccT.
> > >
> > > [3] David Owen. 2024. *How predictable is language model benchmark performance?* arXiv.
> > >
> > > [4] Lukas Finnveden. 2020. *Extrapolating GPT-n performance.* Online.
> > >
> > > [5] 2024. *Predicting emergent abilities with infinite resolution evaluation.* ICLR.
> > >
> > > [6] Sanjeev Arora et al. 2023. *A theory for emergence of complex skills in language models.* arXiv.
> > >
> > > [7] Yangjun Ruan et al. 2024. *Observational scaling laws and the predictability of language model performance.* NeurIPS.
> > >
> > > [8] Jared Kaplan et al. 2020. *Scaling laws for neural language models.* arXiv.
> > >
> > > [9] Jordan Hoffmann et al. 2022. *An empirical analysis of compute-optimal large language model training.* NeurIPS.
> > >
> > > [10] Danny Hernandez et al. 2022. *Scaling laws and interpretability of learning from repeated data.* arXiv.
> > >
> > > [11] Yasaman Bahri et al. 2024. *Explaining neural scaling laws.* PNAS.
> > >
> > > [12] Niklas Muennighoff et al. 2023. *Scaling data-constrained language models.* NeurIPS.
> > >
> > > [13] Berivan Isik et al. 2024. *Scaling laws for downstream task performance of large language models.* ICLR.
> > >
> > > [14] Behrooz Ghorbani et al. 2021. *Scaling laws for neural machine translation.* ICLR.
> > >
> > > [15] Zhang Zhuocheng et al. 2023. *Scaling law for document neural machine translation.* Findings of EMNLP.
> > >
> > > [16] Ethan Caballero et al. 2023. *Broken neural scaling laws.* ICLR.
> > >
> > > [17] Tom Henighan et al. 2020. *Scaling laws for autoregressive generative modeling.* arXiv.
> > >
> > > [18] Usman Anwar et al. 2024. *Foundational challenges in assuring alignment and safety of large language models.* arXiv.
> > >
> > > [19] Mirac Suzgun et al. 2022. *Challenging BIG-Bench tasks and whether chain-of-thought can solve them.* ACL (Findings).
> > >
> > > [20] Jason Wei et al. 2022. *Emergent abilities of large language models.* TMLR.
> > >
> > > [21] Qiyuan Zhang et al. 2024. *Collaborative performance prediction for large language models.* EMNLP.

---

### Official Review · Reviewer_ztqt · 2025-03-14

**Overall Recommendation:** 4

**Summary:**

This paper proposes a novel amortized model-based approach based on Item Response Theory to tackle the problem of the dependence of evaluation procedures on test subset selection and the high cost of running extensive evaluations. Through extensive experiments, the authors show a reduced query complexity while maintaining reliability and better generalization to unseen test subsets.

**Claims And Evidence:**

Claims are properly supported by evidence and design choices are ablated over.

**Essential References Not Discussed:**

None to the best of my knowledge.

**Experimental Designs Or Analyses:**

The experimental design is extensive and sound overall and ablations especially on the use IRT models are very useful for readers not familiar with IRT.

**Methods And Evaluation Criteria:**

The considered methods and evaluation criteria are reasonable/standard and the evaluation setting is extensive.

**Other Comments Or Suggestions:**

- Training details and hyperparameters for the conditional item generator might be better shown as a table in Appendix E.2 + typo in the title for the same section: “Trainig” -> “Training”.

**Other Strengths And Weaknesses:**

- Figures, tables etc aren't properly referenced anywhere throughout the paper, e.g “Figure 2” in line 315 column 2  or “Table 2” in line 809 in Appendix.

**Questions For Authors:**

- How do you make sure that by adaptive testing, the evaluation procedure doesn't overfit some given model?
- Do you have any intuitions about your approach generalizing to a setting where you're training an evaluation model on multiple LLMs (example previous version of some LLM) and using the evaluation model on other LLMs (e.g new iterations of the same LLMs)? A particular example could be the same model's base and instruction-tuned versions.

**Relation To Broader Scientific Literature:**

Getting the most out of evaluations is one of the most crucial problems for properly assessing progress on capabilities and model safety. The authors reduce the cost of evaluations via amortization, while benefits of that can help with generalization to unseen test settings and adaptively tailoring evaluations to the model.

**Theoretical Claims:**

I checked the theoretical claims overall, no issues to be discussed.

---

> ### Author Rebuttal · Authors · 2025-03-28
>
> Dear Reviewer ztqt,
>
> Thank you for your valuable feedback. We answer your comment below.
>
> **Other Strengths And Weaknesses:** Figures, tables etc aren't properly referenced anywhere throughout the paper, e.g “Figure 2” in line 315 column 2 or “Table 2” in line 809 in Appendix.
>
> **Answer:** Thank you for your careful review. We appreciate you pointing out the issues with the referencing of figures and tables. We will correct these references in the latest submission.
>
> **Other Comments Or Suggestions:** Training details and hyperparameters for the conditional item generator might be better shown as a table in Appendix E.2 + typo in the title for the same section: “Trainig” -> “Training”
>
> **Answer:** For the hyperparameters of the conditional item generator, we consistently used a temperature of 0.6, a top_p of 0.9, and a max_tokens of 256. We will update the paper to include these information as you suggest. Additionally, we appreciate you pointing out the typo—we will correct it.
>
> **Questions For Authors 1:** How do you make sure that by adaptive testing, the evaluation procedure doesn't overfit some given model?
>
> **Answer 1:** Thank you for your thoughtful question. Adaptive testing builds on the estimated question difficulty during the calibration phase. We calibrate each question’s difficulty using responses from 172 diverse LLMs, ensuring that the difficulty estimation reflects a broad range of capabilities rather than the trait of any single LLM. This calibration phase identifies which questions are challenging and which are comparatively easier, providing a robust reference. By basing subsequent adaptive testing on these derived difficulty estimates, we ensure that the evaluation process does not overfit a specific LLM.
>
> **Questions For Authors 2:** Do you have any intuitions about your approach generalizing to a setting where you're training an evaluation model on multiple LLMs (example previous version of some LLM) and using the evaluation model on other LLMs (e.g new iterations of the same LLMs)? A particular example could be the same model's base and instruction-tuned versions.
>
> **Answer 2:** Thank you for your question. We calibrate the difficulty of each question by analyzing responses from 172 diverse LLMs—including base versions, instruction-tuned versions, and RLHF versions. This pool spans a wide range of sizes (for example, Llama 3 in 8B, 70B, and 405B, both base and instruct), ensuring that our evaluation framework remains robust and reliable across different training iterations and model sizes, and is applicable to other LLMs. Additionally, we recommend periodically recalibrating the difficulty levels to ensure our evaluation framework remains comprehensive and current with the latest LLMs.

---

### Official Review · Reviewer_ydtz · 2025-03-14

**Overall Recommendation:** 4

**Summary:**

This work proposed a novel way of revisiting large-scale LLM evaluation from IRT perspective. The novel contribution comes from different perspectives: (1) using LLMs to estimate the difficulty of evaluation examples (items), (2) LLM based item generator that can generate a synthetic evaluation example based on the difficulty needed. Using real world benchmarks and pretrained models, they showed how the proposed framework outperformed a strong baseline (model-free classical test theory).

**Claims And Evidence:**

- This work has very strong motivation: number of LLM benchmarks is constantly growing together with the number of models we need to evaluate between each other.
- IRT has a very strong background literature and successful application, but not yet become a standard in LLM evals. This work steps into this direction.

**Essential References Not Discussed:**

n/a

**Experimental Designs Or Analyses:**

I have checked the experimental testbed, and overall they look good to me. Authors provided very thorough description of all experiments and model training for amortized calibration in the appendix.

**Methods And Evaluation Criteria:**

The proposed method is aimed at evaluation of LLMs, and evaluation of this method involves comparison with other frameworks such as CTT. The implemented testbed and selection of LLM models and benchmarks follows the current standard and will be valuable for community.

**Other Comments Or Suggestions:**

n/a

**Other Strengths And Weaknesses:**

n/a

**Questions For Authors:**

Could you please add details about the decoding / sampling parameters that were used (1) in PPO training, (2) during data generation with the trained models? Looks like they were not mentioned in the text.

**Relation To Broader Scientific Literature:**

This work can be very relevant to LLM evaluations community that struggle with the amount of compute and resources needed to keep up with the amount of benchmarks that are required for proper comparisons.

**Theoretical Claims:**

I did not check correctness of proofs given the soundness of empirical experimental results.

---

> ### Author Rebuttal · Authors · 2025-03-28
>
> Dear Reviewer ydtz,
>
> Thank you for your valuable feedback. For both PPO training and data generation, we used a temperature of 0.6, top_p of 0.9, and a max_tokens of 256. We have added this information to the updated submission. We appreciate your attention to detail and hope this clarifies our experimental setup.
>
> Kind regards,
>
> Authors

---

### Decision · Program_Chairs · 2025-05-01

**Decision:**

Accept (poster)

**Comment:**

The reviewers agree that this is an interesting and topical paper that is well written, well motivated and well executed.